# Feasibility of a Real-Time Embedded Hyperspectral Compressive Sensing Imaging System

**DOI:** 10.3390/s22249793

**Published:** 2022-12-13

**Authors:** Olivier Lim, Stéphane Mancini, Mauro Dalla Mura

**Affiliations:** 1University Grenoble Alpes, CNRS, Grenoble INP, TIMA, 38031 Grenoble, France; 2University Grenoble Alpes, CNRS, Grenoble INP, GIPSA-Lab, 38000 Grenoble, France; 3Institut Universitaire de France (IUF), 75231 Paris, France

**Keywords:** compressive sensing, CGNE, DD CASSI, hyperspectral imaging, computation complexity, embedded systems, remote sensing, field-programmable gate array (FPGA), graphics processing unit (GPU)

## Abstract

Hyperspectral imaging has been attracting considerable interest as it provides spectrally rich acquisitions useful in several applications, such as remote sensing, agriculture, astronomy, geology and medicine. Hyperspectral devices based on compressive acquisitions have appeared recently as an alternative to conventional hyperspectral imaging systems and allow for data-sampling with fewer acquisitions than classical imaging techniques, even under the Nyquist rate. However, compressive hyperspectral imaging requires a reconstruction algorithm in order to recover all the data from the raw compressed acquisition. The reconstruction process is one of the limiting factors for the spread of these devices, as it is generally time-consuming and comes with a high computational burden. Algorithmic and material acceleration with embedded and parallel architectures (e.g., GPUs and FPGAs) can considerably speed up image reconstruction, making hyperspectral compressive systems suitable for real-time applications. This paper provides an in-depth analysis of the required performance in terms of computing power, data memory and bandwidth considering a compressive hyperspectral imaging system and a state-of-the-art reconstruction algorithm as an example. The results of the analysis show that real-time application is possible by combining several approaches, namely, exploitation of system matrix sparsity and bandwidth reduction by appropriately tuning data value encoding.

## 1. Introduction

While traditional photography and human vision is based on the acquisition of three spectral bands (i.e., red, green and blue), hyperspectral imaging delivers more precise spectral information as it acquires several tens to hundreds of narrow (e.g., 10 nm-wide spectrally) contiguous bands. Hence, for every sensed spectral band (corresponding to an interval in the light spectrum), information is stored for each point of the scene. As a result, we obtain a data structure called a “hyperspectral cube” or “data cube”. The fine spectral information provided by hyperspectral (HS) imagery has been exploited in various fields such as agronomy to spot crop diseases [1] and in geology to prospect ores [2,3], to cite a few.

Linear scanning is a common acquisition technology for HS imaging [4]. It consists of capturing consecutive slices of the datacube either in the spatial or spectral dimension. However, the large amount of information provided by this imaging technology comes with two major downsides:-Acquisition time: As the whole HS cube is not captured at once, repeated acquisitions are necessary. Depending on the image acquisition mode and the device, it might take even more time to operate the device between every capture step (e.g., changing the filter for the spectral scanning method) than the image acquisition itself. In addition, this acquisition modality cannot handle dynamic scenes.-Data size: due to the high number of spectral bands measured for all spatial pixels, the data cube size increases considerably as the dimensions increase. This amount of data may present a challenge when the imaging device has a limited amount of memory or the acquisitions have to be transmitted through a low-bandwidth channel.

Compressive imagers have been developed to avoid the exhaustive exploration of the data cube required in linear scanning while compressing the data during the acquisition and, thus, overcoming the aforementioned downsides. These devices exploit the Compressive Sensing (CS) theory proposed by Candès et al. [5]. Based on this theory, multiple devices have been developed, such as the Single Pixel Camera (SPC) [6,7] and the Coded Aperture Snapshot Spectral Imager (CASSI) [8] with various variants, e.g., the Double-Disperser Coded Aperture Snapshot Spectral Imager (DD CASSI) [9], the 3D-CASSI [10,11], the Coded Aperture Compressive Temporal Imaging (CACTI) [12] and the Dual-Camera Compressive Hyperspectral Imager (DCCHI) [13].

As scene scanning (either in the spatial or spectral domain) is avoided, the capture time is greatly reduced. However, this reduction of the acquisition time is at the expense, in general, of a loss of spatial or spectral resolution, which should be addressed by computational techniques. Indeed, the raw data acquired by the CASSI cannot be exploited as is, and an inverse problem has to be solved. A large set of algorithms exist to address image reconstruction and recover the data cube, including the Gradient Descent, Orthogonal Matching Pursuit (OMP), Gradient Projection for Sparse Reconstruction (GPSR), Least Absolute Shrinkage and Selection Operator (LASSO) and Iterative Shrinkage/Tresholding (IST); see [14] for a review.

The unavoidable computational operation to reconstruct a datacube usually requires heavy computations. Most of the works in the literature focus on exploring the performance of reconstruction techniques in terms of accurate reconstructions instead of computational analysis. This can be a limitation for applications where real-time hyperspectral analysis is needed (e.g., distinguishing tissues during surgery, such as in [15]). Most of the works in the literature addressing compressive hyperspectral cameras focus more on optical design or on methods for image reconstruction rather than on analysis of the computational complexity of these devices.

The objective of this work is to determine whether an embedded device is able to perform real-time reconstruction. Here, we assume a remote sensor handled by an operator; in this framework, for the definition of “real-time”, we intend a display rate of about 25 frames per second (fps). This rate corresponds to the Phase Alternating Line (PAL) television standard and offers a video stream without noticeable flickering to the operator.

For this purpose, we perform an analysis based on “fast hyperspectral cube reconstruction” technology proposed by Ardi et al. [16]. They exploit the DD CASSI and process the data with the Conjugate Gradient for Normal Equation (CGNE), a convex relaxation algorithm. We made this choice because even if the SPC is a compressive imager, it requires multiple acquisitions, which can be troublesome for real-time applications. In addition, as said in [8], the DD CASSI’s design allows block processing of captured data. Which is beneficial for us since it allows further processing parallelism and may help to reduce the total computation time. Also, it can be noted that the DD CASSI favors spatial information over spectral information when compared to the CASSI but the latter one does not offer block processing.

This paper is a prospective study that brings out ways to accelerate the hyperspectral reconstruction method and determine its performance on an embedded device. This is original in the field of HS imaging in embedded systems, since most of the literature focuses on either HS data compression [17,18] or unmixing [19,20]. Our analysis presents the computational complexity of the method but also includes an estimation of the memory footprint and bandwidth, which are often overlooked in the literature but are crucial when designing a device for real use. These help us outline the specifications of the aforementioned device and determine whether the performance using an state-of-the-art FPGA or heavier computational devices (e.g., GPUs) might be needed or even sufficient for real-time applications. On top of studying the performance, we propose approaches to improve both software and hardware aspects.

This paper is organized as follows. Section 2 introduces the background by presenting snapshot imagers, compressed sensing, and hardware accelerators. Then, Section 3 consists of a theoretical analysis of the CGNE to investigate the expected performance. These performances are presented by using a scene that serves as a study case in order to help the reader understand the CGNE’s requirements. Following that, Section 4 proposes optimizations to achieve real-time reconstruction. The system’s sparsity is first discussed; then, these optimizations start by considering the software part of the CGNE; and finally, we address the hardware implementation of the CGNE and, more notably, the memory and bandwidth used by the CGNE. Section 5 presents experimental simulations on additional study cases to perform a study of the CGNE and then an evaluation of the improvements made by the method presented in this paper.

## 2. Background

This paper addresses the feasibility of a real-time embedded hyperspectral compressive sensing imaging system. This section provides background information on the different fields relevant to make this work self-contained and is organized as follows. First, some hyperspectral imaging devices are introduced to give an overview of some existing acquisition techniques. Then, compressive sensing is discussed. Following that, the acquisition model of the compressive imager used in this work is presented. Finally, hardware accelerators are introduced, as they help speed up the computations thanks to their parallel architecture.

### 2.1. Hyperspectral Imagers

Different types of hyperspectral imagers are available and are based on different acquisition principles [21]. We briefly introduce three different types of snapshot imagers to give an idea of the existing concepts.

The first acquisition modality we present is Spectrally Resolving Detector Arrays (SRDA), which are similar to a Bayer-filter camera. Like a Bayer-filter camera, SRDAs use an array of sensors where each pixel has a filter laid on it. The filters are not limited to green, blue and red but consist of the spectral bands sensed in the desired data cube. This leads to a very compact design that requires less calibration than ones with more optical components. However, the extension in spectral resolution is at the cost of a loss in spatial resolution. Furthermore, such a system is not modular since it binds the spectral resolution, as the filters cannot be modified and are limited to a small number of spectral bands (usually up to around 30).

Another family of hyperspectral imagers is the Snapshot Hyperspectral Imaging Fourier Transform Spectrometer (SHIFT) [22], which acquires an interferogram that can be processed to obtain a hyperspectral image by applying an inverse Fourier transform. The interferograms are produced by using a polarizer, a birefringent prism and a polarization analyzer. The addition of a lenslet array divides the observed scenes into subscenes that go through the system onto the camera sensor. The sub-images undergo different optical difference paths, which allows for a snapshot capture of the scene.

To perform hyperspectral imaging, Gehm et al. [9] proposed a different acquisition scheme based on a device called a Double Disperser Coded Aperture Snapshot Spectral Imager (DD CASSI) that has the particularity of exploiting Compressive Sensing (CS). In order to reconstruct the hyperspectral cube, several acquisitions with different codes are done. These acquisitions are then processed to recover the hyperspectral cube, and the missing data are extrapolated thanks to an a priori model. This system is composed of two dispersive elements (prisms) placed on both sides of a Digital Micromirror Device (DMD). (A Digital Micromirror Device is a microoptoelectromechanical system composed of a matrix of configurable mirrors. Each mirror can be either in a transmission state or a rejection state. In the transmission state, the incoming light is redirected to another part of the optical system, while in the rejection state, the incoming light is redirected to a light absorber or elsewhere.) This also includes a coded aperture and finally a detector (CCD) to capture the two-dimensional multiplexed signals of the three-dimensional data of the scene. The second prism (see Figure 1) balances out the shearing (dispersion and shifting of the light rays) caused by the first prism. This results in measurements that spatially match those of the scene. The data cube projected onto the detector is not sheared; this means that the spectral components of a spatial point in the data cube end up in the same position on the sensor. A pixel of the sensor acquires a mix of the spectral information. A DMD configuration of the coded aperture defines a multiplexing combination. Then, a set of predefined codes enable capturing a subset of mixed spectral information.

### 2.2. Compressive Sensing

Compressive Sensing (CS) is a technique used in signal processing that allows reconstructing of a signal from fewer samples than required by Nyquist–Shannon sampling theorem. Its principle is based on exploiting sparsity in a suitable representation domain for the signal X we seek to recover. The signal $X\in R^{N}$ can be represented in an orthonormal basis Ψ of dimension *N* with *S* coefficients, where S<<N.

In practice, the set of *M* measurements is made in a second basis Φ that should not be correlated with Ψ. The measurements Y are then obtained as:(1)$$Y=\Phi X=\Phi \Psi \theta =\Phi \sum_{s=1}^{S}\theta_{s}\psi_{s}$$
where Y is a vector of size *M* and Φ the measurement matrix of size M×N, where M<N. Ψ is of size N×N with $\psi_{s}$ as columns, and θ is an *N* vector with *S* non-zero elements that form the coefficients. X is then a linear combination of *S* vectors.

By encoding X in Φ, we obtain compressed data of this signal. This compression reduces transmission and storage requirements. CS can be understood as the acquisition of a subset of mixtures of a signal X, and reconstruction is equivalent to untangling the mixed signals to recover X. The missing information is recovered thanks to the prior knowledge encoded as a regularization function, for instance. Recovering from (Equation 1), and since M<N, gives us an inverse problem that can be solved as an $\ell_{1}$-optimization problem:(2)$$\hat{\theta }=\arg\min_{\theta }{∥\theta ∥}_{1}\;\mathrm{such}\mathrm{that}Y=\Phi \Psi \hat{\theta }$$

To solve the optimization problem (Equation 2), several algorithms have been presented (see [23]). Among these, convex relaxation algorithms and greedy pursuit are the most popular in the literature due to their implementation simplicity while being quite efficient. Regarding convex relaxation algorithms, well-known algorithms are the Gradient Descent, the Basis Pursuit [24] and the Least Absolute Shrinkage and Selection Operator [25]. For greedy pursuit, Orthogonal Matching Pursuit [26], Stage-wise Orthogonal Matching Pursuit [27], Iterative Hard Tresholding [28] and Fast Iterative Shrinkage/Thresholding Algorithm (FISTA) [29] are often mentioned in the literature.

More specific to the hyperspectral compressive sensing framework, Total Variation (TV), Gradient Projection for Sparse Reconstruction (GPSR) [30], the Two-Step Iterative Shrinkage/Thresholding (TwIST) [31] and Alternating Direction Method of Multipliers (ADMM) [32] are widely used.

Alongside the aforementioned image reconstruction techniques, the recent emergence of artificial intelligence has led to deep-learning-based algorithms offering a novel alternative to reconstruct the data cube. See [33] for a discussion of both the optimization and deep learning approaches. These deep learning algorithms offer satisfying performance; e.g., [34] reported 30 frames per second for a 256×256×24 scene, but they require a training phase that can last hours or even days and needs a training dataset.

Finally, when reconstructing the data cube, the algorithm is not the only parameter to consider. For example Liu et al. [35] incorporate rank-minimization in their proposed reconstruction method; similarly, Zha et al. [36] do the same with Group Sparse Representation (GSR) [37] but also combine both the FISTA and ADMM; Zhang et al. [38] choose to mix ADMM and the TWiST while using a dual-camera, the DCCHI. Additionally, a domain transform can also be used; e.g., Ma et al. [39] propose a deep-neural network to exploit the Fourier Transform, the ADMM and low-rank tensors.

### 2.3. DD CASSI Matrix Model

In this section, we present the principle of the DD CASSI, which is the compressive hyperspectral imaging device that we have chosen as the example to consider in this work.

We follow the work of Ardi et al. [16], who proposed a convex relaxation algorithm to recover the HS data cube. Although the DD CASSI is a snapshot imager, Ardi et al. [16] suggested performing the cube recovery based on multiple acquisitions. This improves the reconstruction quality (see Section 5.2.2).
Matrix modeling of the DD CASSI acquisitions 


The objective is to reconstruct the scene o from the acquired data d and the measurement matrix H. Ardi et al. [16] modeled scene acquisition as:
(3)$$d=\mathrm{Ho}\;\mathrm{where}\;d=\left[\begin{array}{c} I^{\left(1\right)}\\ \vdots \\ I^{\left(N\right)} \end{array}\right]\;\mathrm{and}\;H=\left[\begin{array}{c} T^{\left(1\right)}\\ \vdots \\ T^{\left(N\right)} \end{array}\right]$$
where d is a concatenation of the *N* acquisitions I, T describes the DMD configurations for each acquisition, H represents the optical transform and filtering induced by the system, and o are the data of the observed scene.

Let *R*, *C* and *W* be the dimensions of o, i.e., *R* rows, *C* columns and *W* wavelengths. Then, d is of size NRC, each acquisition I has a size of RC, each T is a matrix of dimension RC×RCW and H is a matrix of size NRC×RCW.


Hyperspectral cube recovery 


Reconstructing the scene o from (Equation 3) is an ill-posed problem since the solution is not unique. To solve this problem, it is possible to resort to Tikhonov regularization. Hence, the regularized reconstruction problem becomes:(4)$$\hat{o}=\arg\min_{o}{\{∥d-\mathrm{Ho}∥}^{2}+\Omega \left(o\right)\}$$
with $\hat{o}$ as the estimation of o, $\Omega \left(o\right)=\mu_{x}∥D_{x}o{∥}^{2}+\mu_{y}∥D_{y}o{∥}^{2}+\mu_{\lambda }{∥D_{\lambda }o∥}^{2}$ as the penalization function, where $D_{x}$, $D_{y}$ and $D_{\lambda }$ are, respectively, the finite differences along the spatial dimensions *x*, *y* and the spectral dimension λ and their associated regularization coefficients. Here, regularization favors continuous data.

As the function in (Equation 4) is convex and differentiable, it admits an analytical solution:(5)$$\hat{o}={\left(H^{⊤}H+\mu_{x}{D_{x}}^{⊤}D_{x}+\mu_{y}{D_{y}}^{⊤}D_{y}+\mu_{\lambda }{D_{\lambda }}^{⊤}D_{\lambda }\right)}^{-1}H^{⊤}d$$

However, due to the size of the matrices, this solution is impractical to compute directly in real scenarios of applications. Thus, to reconstruct the data cube, it is possible to use an iterative method such as the Conjugate Gradient for Normal Equation (CGNE) [40], given in Algorithm 1, and rewrite the problem as:(6)$$M^{⊤}M\hat{o}=H^{⊤}d$$
with $M=[H,\sqrt{\mu_{x}}D_{x},\sqrt{\mu_{y}}D_{y},\sqrt{\mu_{\lambda }}D_{\lambda }]$. The CGNE is used to solve problems of form Ax=b. In our case, $A=M^{⊤}M$ and $b=H^{⊤}d$.
**Algorithm 1:** CGNE algorithm. Convergence condition is reached when a certain precision tol is met and can be evaluated as $r_{i}^{⊤}r_{i}<tol$.
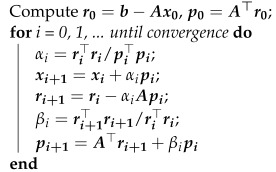


### 2.4. HW Accelerators

Since solving (Equation 2) can be a computationally demanding task, hardware (HW) accelerators can be used to achieve real-time reconstruction. Central Processing Units (CPUs) are considered the brain of a computer, but they are not as efficient as specialized chips, notably when parallelism and computing power are required. In this case, HW accelerators such as Graphics Processing Units (GPUs) or Field Programmable Gate Arrays (FPGAs) are better choices.

GPUs’ initial purpose is to render digital pictures and videos. They are very efficient at executing a large number of arithmetic operations in parallel. This is done by having a massive amount of computing units organized in repeated “grid” patterns along with memory caches to reduce the latency. This amount of resources comes with significant power consumption (e.g., 350W for an Nvidia 3080 [41] and 230W for an AMD RX 6700XT [42]).

FPGAs are reprogrammable integrated circuits. They offer parallelism at a finer level, and configuring them makes them versatile while being energy-efficient. They incorporate memory components in their fabric, which minimizes the latency and makes FPGAs great solutions for real-time applications.

Both GPUs and FPGAs are designed to perform a multiply–accumulate (MAC) operation, which is the product of two variables added to an accumulator: a←a+(b×c). Hence, they are very efficient at doing matrix–vector or vector–vector products since they are composed of only MAC operations.

Regardless of the device, we can distinguish two types of memory: “work memory” and “global memory”. Work memory is located closer to the computing units to offer fast data access but is of a limited amount, whereas the role of the global memory is to store all the required data but at the cost of reduced speed. When the data set is larger than the work memory, the system swaps data and copies them from global to work memory to perform computations. Hence, the size of the work memory is crucial regarding the bandwidth since it determines whether swapping is needed or not.

A description of GPU memory hierarchy is available at [43]. For GPUs, since L1 caches get invalidated between each computation task, we will consider that the L2 cache acts as work memory. Their global memory is made of memory chips located outside the GPU die but on the same package. For FPGAs, the work memory is the embedded memory laid between the circuit blocks, and the global memory is external to the device, and its access must be done through a memory controller.

## 3. Naive CGNE Implementation: Performance and Limits

This section presents the motivation of this work. The contribution of this section dwells in the theoretical analysis of the CGNE’s initial specifications regarding computing power, memory footprint and data bandwidth. However, by estimating the performance on an FPGA or a GPU, it is shown that a reconstruction would take several seconds at best. Thus, optimization is required to achieve real-time reconstruction.

This section is organized as follows. First, we introduce the two reconstruction models we considered during this paper. Then, we evaluate CGNE’s initial specifications. After that, we introduce a model GPU and a model FPGA with their respective performance. This will help the reader to get a better grasp of the magnitude of the CGNE’s specifications. However, we see that they far exceed the capabilities of the computing devices.

### 3.1. Reconstruction Models

From the matrix model described in Section 2.3, two reconstruction models can be used:Multi-row model (MR): from [16], the model described in Section 2.3. This model takes into account correlation between *x*, *y* and λ.One-row model (OR): from [44]. Each row of the observed scene is reconstructed one-by-one and then concatenated together to form the restored scene. During the reconstruction, only correlations along *x* (pixels in the same row) or λ (different wavelengths) can be taken into account through regularization parameters $\mu_{x}$ and $\mu_{\lambda }$.

In the OR model, the reconstructions are performed on one row instead of on the whole scene as in the MR model. This results in smaller dimensions (CW instead of RCW) for the vectors and matrices; e.g., H now contains the DMD configuration of one row instead of the whole scene. This reduces the memory footprint of the CGNE, which is advantageous for implementations on embedded devices. In addition, the term $\mu_{y}{∥D_{y}o∥}^{2}$ is removed from Ω(o), $\mu_{y}{D_{y}}^{⊤}D_{y}$ is removed from Equation 5, and $\sqrt{\mu_{y}}D_{y}$ is removed from M.

### 3.2. CGNE Performance Study

#### 3.2.1. CGNE Algorithm Computation Cost

Since the size of the matrices and vectors are known, we can determine the computing cost of the CGNE (see Algorithm 1). We recall that A is RCW×RCW; then, $x_{i}$, b, $r_{i}$, $p_{i}$ are RCW vectors; and finally, $\alpha_{i}$ and $\beta_{i}$ are scalars. Note that for iteration *i*, computing $r_{i+1}^{⊤}r_{i+1}$ and storing the result spares us the computation of ${r_{i}}^{⊤}r_{i}$ in Steps 3 and 6.

Let ITER be the total number of CGNE for-loop iterations. Then, the total computation cost for a scene for the OR model is given by:(7)$$R[2{\left(CW\right)}^{2}+ITER\left(2{\left(CW\right)}^{2}+5CW\right)]$$

The total computation cost for a scene for the MR model is given by:(8)$$2{\left(RCW\right)}^{2}+ITER\left(2{\left(RCW\right)}^{2}+5RCW\right)$$

Using the MR model, the computation amount is multiplied by *R* compared to the OR model. This difference is due to the fact that the *y* dimension has an exponential impact in the MR model (directly incorporated in H; see Section 4.1.1).

#### 3.2.2. Memory and Bandwidth

Now that the required computation power has been calculated, we focus on the memory footprint and bandwidth. We start by determining the memory footprint used by running the CGNE algorithm then present a basic estimation of the bandwidth.


Memory footprint 


Looking at the CGNE (see Algorithm 1), we have to store its variables along with intermediate ones to run all the computations. When looking closely at the CGNE, we notice that it is not necessary to store all these variables. Indeed, $x_{i+1}$, $p_{i+1}$ and $r_{i}$ can be stored in the same memory unit used by $x_{i}$, $p_{i}$ and b, respectively, since $x_{i}$ and $p_{i}$ are not used after $x_{i+1}$ and $p_{i+1}$ affectations, and b is only used to initialize $r_{0}$. Additionally, regarding intermediate variables, allocating an RCW vector is sufficient. We recall that we also store $r_{i+1}^{⊤}r_{i+1}$, just as discussed in the previous section. Hence, in total, we have to store
(9)$${\left(RCW\right)}^{2}+4RCW+3\;\mathrm{values}$$

Since the video consists of multiple reconstructions per second, the memory footprint turns into bandwidth. Without memory swapping, the bandwidth for a video is obtained by multiplying the memory required for the inputs and outputs of the CGNE by the target number of frames per second. When swapping is required, the worst case consists of storing and loading from the global memory every variable for every iteration of the CGNE, since they do not fit into the work memory. Hence, swapping might tremendously increase the bandwidth.

#### 3.2.3. Theoretical CGNE Performance in a Study Case

Since we know the theoretical specifications of the CGNE, we compare them with the specifications of a GPU and an FPGA in order to determine the reconstruction rate in a naive setting. For the GPU, we chose the NVIDIA RTX 3090 Ti [45], and Xilinx Ultrascale + VU13P [46] is the FPGA. They are both (at the time of writing) very powerful devices in their respective categories. Their specifications are given in Table 1. The performance of the CGNE is evaluated for the scene depicted in Figure 5a. This is a 30×150×31 scene. Although its dimensions are quite modest, the specifications of the CGNE for this scene are already very demanding.

Each reconstruction of our study case requires 1.99 TMAC for the OR model and 59.55 TMAC for the MR model (see (Equation 7) and (Equation 8)). (TMAC stands for TeraMAC and is equal to $10^{1}2$MAC. We recall the International System of Units prefixes: Peta (P: $10^{15}$), Tera (T: $10^{12}$), Giga (G: $10^{9}$), Mega (M: $10^{6}$) and Kilo (k: $10^{3}$).) Considering the computation cost, the OR model would achieve 10 reconstructions per second for the RTX 3090 Ti and 5 reconstructions per second for the VU13P. Note that since the computation cost depends on ITER, the number of CGNE for-loop iterations, we evaluated the total computation cost for the average value we obtained during the simulations (see Section 5).

Regarding memory footprint, let us consider every value is stored in 64 bits; from (Equation 9), we deduce that it reaches 173 MB per row of the OR model, 5.2 GB in total for the OR model and 156 GB for the MR model. Since the memory footprint exceeds the work memory for both devices, memory swapping is needed (see Section 4.3.3). Therefore, the memory transfer for a reconstruction would be several TBs for the OR model and hundreds of TBs for the MR model. Because of this, reconstruction with the OR model would take around 15 s for the RTX 3090 Ti and 125 s the VU13P. These reconstruction times are not suitable for real-time applications, and we have to find ways to reduce them.

As we saw through this section, the CGNE’s specifications far exceed the resources provided by the selected computing devices, and no single device can fulfill these requirements. However, in the next sections, we describe the structure of the matrix A, especially its sparsity, and then we explain how we will exploit it to improve the performance of the CGNE and achieve real-time reconstruction.

## 4. CGNE Optimization for Real-Time

This section presents the techniques we used to reduce the CGNE specifications and achieve real-time performance. This is done by using a sparse matrix format and finely tuning the representation of data. The contribution is the study of the sparsity of matrix A and the theoretical analysis of the improved computational cost, memory footprint and data bandwidth.

In this section, we first explain how H is built; then, we determine where the non-zero entries are in A, and, more importantly, we show that they amount less than 1% of A’s entries. Then, a sparse matrix format is presented and is used in order to improve the performance of the CGNE.

Following that we explain how to save both bandwidth and hardware resources by finely tuning the representation of data. Thus, we briefly introduce fixed-point data value encoding and explain how to determine the data format. In conclusion, we show that several thousands reconstructions per second are possible for both devices with the OR model. Regarding the MR model, the VU13P is capable of several thousands of reconstructions per second, whereas the RTX 3090 Ti is limited to a dozen reconstructions because of memory swapping.

### 4.1. System Sparsity

#### 4.1.1. H Construction and Subsequent A Structure


One-row model 


In order to exploit the sparsity of H and reduce the number of operations, we need to identify the structure of its non-zero entries. For that, we rely on both the optical system it models and on the multiple acquisition procedure incorporated in H. We recall that d contains the concatenation of the set of acquisitions of the camera sensor. The matrix H reproduces the co-location property of the optical system; that is, a pixel of the sensor perceives a spectral mixture of components from the same spatial position. Hence for each different wavelength that belongs to the datacube, H is composed of a diagonal line. The diagonal lines are made of the DMD configurations. As we go through the wavelengths, the diagonal is shifted along one direction or another to match the linear dispersion caused by the prism. To model multiple acquisitions with different codes (DMD configuration), this pattern is replicated and concatenated vertically for each acquisition, as shown in Figure 2a.

As a result, A is symmetric and shows diagonal entries that start at columns 0, 1 and wC with 1≤w≤W; see Figure 2b. Thus, for our study case where W=31, a given line of A has at most 33 nonzero entries and 4617 zero values for a fully open DMD. Note that the sparsity of A depends on the DMD configuration.

Now that the arrangement of nonzero entries in A is known, we can estimate their number, which we denote as $E_{OR}$. For a square matrix of size *n*, a diagonal at position *i* has a length of n−i. We can deduce that:(10)$$E_{OR}=CW+2\left(CW-1\right)+2\left(\sum_{i=w}^{W-1}CW-wC\right)=\left(W+2\right)CW-2$$

The matrix density (i.e., the number of nonzero entries) is related to the matrix size and is expressed as $\frac{E_{OR}}{{\left(CW\right)}^{2}}$. In our study case, it is equal to 0.7%.


Multi-row model 


Because the MR model takes into account the relation between a set of rows to differentiate the acquisitions, the different rows and the distinct wavelengths, the previous pattern for the OR model is replicated and shifted along the horizontal and vertical axes, as shown in Figure 3a. Then, in Figure 3b, we can see that A contains a repetition of OR’s A pattern with additional diagonals. Generally speaking, diagonals are located at positions 0, ±1, ±RC, and *R* distinct square submatrices of size CW are placed along the main diagonal. These submatrices contain W−2 diagonals placed at positions ±wC with 2≤w<W. We can also notice that in the diagonals at positions ±1, every iC-th entry with i∈[1,RW−1] is a 0.

Following the same approach used to estimate the number of entries in OR’s A, we conclude that $E_{MR}$ is, at most, equal to:(11)$$\begin{array}{cc} E_{MR} & =RCW+2\left(RCW-1\right)+2\left(n-C\right)+2\left(n-RC\right)+2R\left(\sum_{w=2}^{W-1}CW-wC\right)-2\left(RW-1\right)\\ \; & =RCW^{2}+2\left(2RCW-RW-C\right) \end{array}$$

Density of A is $\frac{E_{MR}}{{\left(RCW\right)}^{2}}$, which is equal to 0.003% for our study case.

#### 4.1.2. Sparse Matrix Format

As discussed previously, the matrix A is very sparse (and especially in the MR model). Exploiting this property will be beneficial, since a sparse matrix format improves both the storing space and the computational cost of arithmetic operations. Furthermore, as the CGNE relies on matrix–vector multiplications, exploiting the sparse matrix formats advantageously reduces the computing time.

Taking into account the patterns of entries in A and the matrix–vector multiplication operation, two sparse matrix formats are of interest: Compressed Sparse Row (CSR) and Diagonal format (DIA); both are described in [48]. However, because the DIA format stores any diagonal having at least one nonzero entry and the potential sparsity of the diagonals of A, the CSR format (described in Figure 4) might be more suitable to store the A matrix.

The CSR format and its associated matrix–vector multiplication algorithm (cf. Algorithm 2) guarantees that no iteration over zero entries will be done, and for each nonzero entry in A, only addition and multiplication are performed.

By using this format, storing A uses $E\times \left(N_{b}+\left[log_{2}\left(RCW\right)\right]\right)+\left(RCW+1\right)\times \left[log_{2}\left(E\right)\right]$ bits, where $N_{b}$ denotes the number of bits used to encode an entry of A (e.g., 3, 16, 19, 32 or 64 bits).
**Algorithm 2:** Matrix–vector multiplication using CSR format: *y* and *x* are, respectively, the result vector and the operand vector.
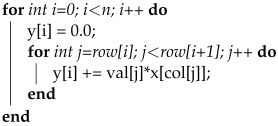


### 4.2. Software Acceleration of Compressed CGNE

In this section, we determine the computational cost of the CGNE exploiting the sparsity of A. We reduce the number of computations by using a sparse matrix format, which allows us to reach several hundreds of reconstructions per second.

#### Theoretical Performance Evaluation

As discussed in Section 4.1.2, the matrix A contains less than 1% nonzero entries, favoring a significant improvement by using the CSR format and getting rid of unnecessary operations in matrix–vector products. We can then count the number of MAC and arithmetic operations for each step of the CGNE, as shown in Table 2. Then, for the OR model, the new computing cost ${\mathrm{COS}T}_{\mathrm{OR}}$ for a scene is:(12)$$\begin{array}{cc} {\mathrm{COS}T}_{\mathrm{OR}} & =R\times \{2E_{OR}+ITER\times \left[2E_{OR}+5CW\right]\}\\ \; & =R\times \{2((W+2)CW-2)+ITER\times [2((W+2)CW-2)+5CW]\}\\ \; & =R\times \{2\left(CW^{2}+2CW-2\right)+ITER\times \left[2CW^{2}+9CW-4\right]\} \end{array}$$

The computation cost of a scene with the MR model ${\mathrm{COS}T}_{\mathrm{MR}}$ is:(13)$$\begin{array}{cc} {\mathrm{COS}T}_{\mathrm{MR}}= & \;2E_{MR}+ITER\times \left[2E_{MR}+5RCW\right]\\ = & \;2(RCW^{2}+2\left(2RCW-RW-C\right))+\\ \; & +ITER\times [2\left(RCW^{2}+2\left(2RCW-RW-C\right)\right)+5RCW]\\ = & \;2\left(RCW^{2}+2\left(2RCW-RW-C\right)\right)+ITER\times \left[2RCW^{2}+13RCW-4RW-2C\right] \end{array}$$

From Table 2, we can deduce the number of MAC per second required for a reconstruction and then the maximum reconstruction rate achievable. For our study case, we reported these figures in Table 3. Note that scalar operations are ignored since they are negligible. By using the CSR format, it is possible to reach several hundreds of reconstructions per second, whereas the naive implementation only allows fewer than 10. We can also notice that using the sparse matrix format makes the computational costs of the OR and MR models quite similar, 15.15 GMAC and 16.00 GMAC, respectively, while it is 1.99 TMAC and 59.55 TMAC for the naive implementation.

However, the computing power is not the only limiting factor, and memory footprint and bandwidth still need to be considered. This will be the topic in the next section.

### 4.3. FPGA Hardware Implementation of Optimized CGNE

#### 4.3.1. Fixed-Point Data Value Encoding

Fixed-point data value encoding uses a fixed number of bits for both the decimal and fractional parts of values. In short, fixed-point offers some advantages over floating-point: better flexibility in choosing the data size, well-supported by FPGAs, requires fewer logic gates, and has lower latency and lower power consumption. However, the drawback is the reduced precision, and since floating-point encoding uses an exponent, fixed-point’s range is narrower than floating-point’s and the step between two values is also larger for fixed-point. Further information on fixed-point encoding is available in [49].

#### 4.3.2. Computational Noise and Memory Footprint

The precision losses and reiterating computations with inaccurate data leads to computational noise and may affect the results of the CGNE. As we use fixed-point encoding to minimize the memory footprint, we have to study the effect of computational noise, but this also has to be studied with 32-bit floating-point (FP32) since the RTX 3090 Ti uses this encoding for its computing units. Then to study the effect of computational noise, the CGNE is run with different levels of precision, and the reconstruction quality is evaluated using image-quality metrics.

Using this method, we determine the memory footprint of the CGNE using either fixed-point or FP32 encoding. The results are summed up in Table 4. The experiment is detailed in Section 5.3. As a conclusion, we can notice that it is possible to cut down the memory footprint by half.

Considering data bandwidth, the performance depends on whether the memory footprint fits into the work memory of the computing device. If not, memory swapping is activated, which increases the data bandwidth. We go into detail in Section 4.3.3.

#### 4.3.3. Bandwidth Analysis


Non-swap approach 


This case happens when the memory footprint does not exceed the work memory. This is particularly the case of the OR model, since scene reconstruction is done row-per-row and the variables use less memory.

Determining the required bandwidth in this case is straightforward. For each row, we need to load the inputs (*A* and *b*) and then store the output (*x*) at the end of the CGNE. The bandwidth is the product of the row’s footprint times the number of rows in the scene and the video framerate (in our study case, 25). Considering the data encoding chosen after computational noise of Section 5.3, we sum up in Table 5 the required bandwidth for our study case, and the maximum rate of scene reconstruction we can attain based on the devices’ bandwidths. Looking at these numbers, by using the OR, both the RTX 3090 Ti and VU13P have the required bandwidth to perform 25 reconstructions per second. In complement, their bandwidth allows attainment of several thousands of reconstructions per second; in that case, the limit comes from the computing power.


Swap approach 


Memory swapping is activated when the memory footprint exceeds the work memory. Variables are loaded into work memory when needed and unloaded afterwards to free up memory space. This greatly increases the bandwidth, as a given variable can be loaded multiple times during the CGNE. In our study case, swapping is activated for the MR model with the RTX 3090 Ti, whereas the VU13P’s work memory is higher than the data footprint.

The bandwidth estimation depends on how many times a variable is used throughout the CGNE and its memory size. Because of the iterative nature of CGNE, some data may be re-used between the algorithm’s steps and do not require to be reloaded every time. This results in the bandwidths and the associated maximum fps values given in Table 6.

Studying these numbers, we see the CGNE’s bandwidth running on the RTX 3090 Ti is more than three times higher than on the VU13P, while the footprint is lower (see Table 4). This is due to the lower amount of work memory available in the RTX 3090 Ti, which leads to swap activation. However, the RTX 3090 Ti is able to perform 12 reconstructions per second, and the VU13P reaches a few thousands reconstructions per second.

## 4.4. CGNE Feasibility

As a conclusion of this section, both appropriately tuned fixed-point and FP32 encodings reduce the memory footprint by half without significantly affecting the reconstruction quality (see the experiments presented in Section 5.3 for details). As a result, the RTX 3090 Ti and the VU13P both have enough bandwidth to perform several thousands of reconstructions per second with the OR model. With the MR model, only the VU13P is still able to attain real-time (25 fps); the RTX 3090 Ti can only achieve a dozen reconstructions per second because of memory swapping.

## 5. Experiments

This section presents the experiments performed to evaluate the method we proposed in previous sections. First, the method for the experiments is described. Then, an experimental study of the CGNE itself explores the reconstruction quality, which has logarithmic growth with regard to the number of acquisitions and the number of iterations. Afterwards, through a simulation run on a CPU, we assess the improvement provided by sparse matrix formats and show that the reconstruction time is reduced by several hundredfold. Finally, simulations of computational noise are completed to confirm that reduced precision of data encoding can be used without impacting the reconstruction quality. This asserts that the CGNE can be run on both the RTX 3090 Ti and the VU13P, and, more generally, on GPUs and FPGAs.

### 5.1. Method

In order to validate the reconstruction algorithm and evaluate the importance of several of the parameters affecting the quality of the reconstruction, experiments are conducted on the scenes presented in Figure 5.

Quality metrics to evaluate the reconstruction quality are mandatory since visual inspection is not conclusive. Structural Similarity (SSIM) and Peak Signal-to-Noise Ratio (PSNR) are used hereinafter. They are measured relative to the reference hyperspectral data cube. We recall that the PSNR expresses the ratio between the maximum power of a signal and the power of the error (or noise) after reconstruction. SSIM is used to measure similarity between an image and its reference. A description of the PSNR and SSIM is given in [50].

Note that since finding good regularization parameters $\mu_{x}$, $\mu_{\lambda }$ and $p_{tk}$ can be tedious for each scene, we started with the values provided in [16] and then found the ones that provided the best reconstruction quality and the lowest number of CGNE iterations. For each scene, the values of these parameters are given in Figure 5. Then, for the CGNE convergence condition (cf. Algorithm 1), $tol=10^{-6}$.

**Figure 5 sensors-22-09793-f005:**
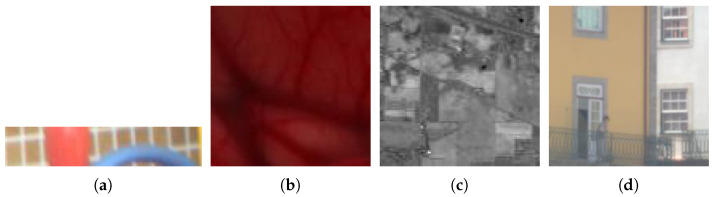
Scenes used in the experiments along with their size, their source and the values of ($\mu_{x}$$\mu_{\lambda }$, $p_{tk}$). (**a**) “Toys” (30×150×31) [51]—(5, 0.5, 0.1). (**b**) “Brain” (100×100×100) [52]—(5, 0.5, 0.1). (**c**) “Indian pine” (100×100×100) [53]—(5, 0.5, 0.1). (**d**) “Urban” (200×200×33) [54]—(5, 0.7, 0.7).

### 5.2. Analysis of CGNE Quality

#### 5.2.1. Quality Evolution over Iterations

The evolution of the reconstructed image quality as a function of the number of iterations of the CGNE algorithm is measured by the “Time to Quality” (TQ), that is, the number of iterations to reach 95% of the best quality, i.e., the maximum value for a given metric. We denote ITER the number of iterations before the CGNE reaches convergence for a given line, and $ITER_{min}$ is the lowest ITER for a given scene, i.e., the lowest ITER among all lines. The quality is averaged over all the lines of the scene and measured at each iteration of the CGNE. Since the CGNE converges in a different ITER for each line, the evolution of the quality is depicted up to $ITER_{min}$. For all scenes, the evolution of the quality is depicted in Figure 6 and we can see that TQ is reached at around 30% of $ITER_{min}$. We can also notice that the reconstruction quality follows logarithmic growth: the CGNE algorithm performs most of the reconstruction in the early iterations (i.e., the first 25% of TQ), and then small improvements in quality are made in further iterations.

We can conclude that the CGNE does not perform linearly with regard to the number of iterations but, more importantly, setting a strict tolerance condition can lead to high computation times without significantly increasing the reconstruction quality. In other words, it is possible to trade a slight reduction in image quality with a significant saving in computation time.

#### 5.2.2. Impact of the Number of Acquisitions on Quality

The influence of the number of acquisitions, *N*, is studied by evaluating the PSNR and SSIM for a set of values of *N*. As expected, the quality improves when *N* increases because there is more and more information from which to reconstruct the data cube. Since the scenes have different spectral dimensions, we compared the measures while considering the Ratio of Captured Information (RCI), i.e., N/W, rather than *N* solely.

Figure 7 shows that both the PSNR and SSIM do not increase linearly with regard to *N* but have logarithmic growth instead.

The SSIM rises sharply and quickly stalls when RCI comes to about 0.25 but reaches a high value in the end: for an RCI of 1, the worst SSIM is equal to 0.91 for the Brain scene and the best is 0.99 for the Urban scene.

Regarding the PSNR, it does not show any stalling, but the maximum curvature is reached at around an RCI of 0.25, which means that over this point, additional acquisitions improve relatively less than previous ones.

### 5.3. Computational Noise Experiments

To simulate the computational noise in fixed-point encoding, the precision loss on the fractional part of the data representation is estimated by converting numbers to their equivalent fixed-point rounding of negative power of 2. For the fixed-point encoding, the simulations are run with different numbers of bits for the fractional part. This number is denoted *v*. Using different values of *v* helps us to determine the lowest threshold of precision, and thus of memory footprint, that we can afford without degrading the reconstruction quality.

In order to distinguish the computational noise from the intrinsic reconstruction noise, reconstruction simulations are conducted for several numbers of acquisitions up to an RCI of 1.0, as in Section 5.2.2. Because it offers the best precision, 64-bit floating-point encoding (FP64) is used as the reference to compare the other encodings. In addition, to keep a fair comparison between data representations, for a given line, the reconstructions are stopped after a fixed number of iterations instead of the fulfillment of the convergence condition. This limit is defined by the number of iterations reached during the FP64 reconstruction, which is stopped when the convergence condition is fulfilled. The results are depicted in Figure 8.

The FP64, 32-bit floating point (FP32) and fixed-point representations with v={32,24} have the same reconstruction quality, since they overlap on Figure 8. So using any of these number representations does not hinder the reconstruction quality.

We can also consider that v=16 offers acceptable precision. Although its PSNR is lower by 2.9 dB for the “Brain” scene at RCI = 0.05, the difference shrinks to less than 1 dB for all scenes starting from RCI = 0.1, and then shows very little difference from FP64 for higher values of RCI.

Regarding v={8,12}, they can definitively be discarded, as they do not offer decent quality. For v=12 and RCI = 1.0, the scenes “Brain”, “Indian pine” and “Urban” show a difference of, respectively, 4.58, 8.47 and 10.68 dB.

### 5.4. Sparse Matrix Format Experiments on CPU

To assess the improvement of sparse matrix formats on **A**, the reconstruction times when using them are compared with the reconstruction times when using the array format, i.e., the default format. This comparison is made with the following sparse formats: Block Sparse Row (BSR), COOrdinate (COO), Compressed Sparse Column (CSC), Compressed Sparse Row (CSR) and DIAgonal (DIA). These formats are described in [48].

The comparison is based on the average improvement factor offered by the sparse matrix formats. That is, for each row of a scene, on average, how much is the reconstruction time divided when using a sparse matrix format. Figure 9 depicts the resulting improvement factor for each scene.

We can see that the the CSR format offers the best improvement and reduces the reconstruction time by several hundredfold. At worst, the DIA format reduces the reconstruction time by several tenfold. This experiment confirms the gain from using sparse matrix formats.

### 5.5. Estimated Performance of Optimized CGNE

In this section, we present the computing power and bandwidth used by the CGNE algorithms for 25-fps reconstructions. In order to give a broader view, we consider further scene resolutions as well as the ones presented in Figure 5. As we focus on the OR, and for readability reasons, the performances are given only for this model.

We recall that performance depends on the number of CGNE iterations. We denote $ITER_{perf}$ the number of iterations used to estimate the specifications and $ITER_{sim}$ the number of iterations reached during the simulations. For scenes presented in Figure 5, their performances were estimated while considering $ITER_{perf}$ equal to the average $ITER_{sim}$. For the additional resolutions, as we cannot run simulations for them, $ITER_{perf}$ is considered as $\frac{1}{2}CW$. This choice is made because $\frac{1}{2}CW$ is half the dimension of the problem we seek to solve with the CGNE (i.e., reconstruct *x*). Additionally, the simulations do not provide any conclusions on the value of $ITER_{sim}$ with regard to the scene’s dimensions.

Figure 10 and Figure 11 depict, respectively, the computing power and the bandwidth for the Nvidia RTX 3090 Ti and Xilinx VU13P. In these figures, “Naive” refers to a basic implementation of the CGNE, “Sparse” is the implementation with the sparse matrix format (cf. Section 4.2), and “HW” enables fixed-point and 32-bit floating-point (cf. Section 4.3). On each Figure, the filled areas depict the RTX 3090 Ti’s and the VU13P’s rated performances regarding either the computing power or the bandwidth performance. Note that the “HW” implementation is not present in Figure 10 since the total number of operations remains the same. Further note that the scenes “Brain” and “Indian Pine” have close values regarding $ITER_{sim}$, and because these scenes have the same resolution, their specifications are nearly identical and overlap in the aforementioned figures.

By studying the results, we see that implementing the SW acceleration with the sparse matrix format divides both metrics by several hundredfold to a hundred thousandfold at most. Such dramatic improvements are reached by careful exploitation of the sparsity of the system matrix *A*, which enables us to skip huge amounts of multiplication by zero and saves memory footprint by not storing 0 entries.

HW acceleration provides a similar magnitude of enhancement for the bandwidth when the required memory footprint is just at the border between the “non-swap” and “swap” modes (see Section 4.3.3). This is the case for the RTX 3090 Ti with scenes “Brain” and “Indian pine”. In these cases, without the HW acceleration, the reconstruction rate is lower than 1 fps because of the bandwidth limitation. When memory swapping is not needed, HW acceleration halves the memory footprint and, thus, the bandwidth. Final reconstruction rates for the simulated scenes are reported in Table 7.

As a brief comment regarding the MR, the reconstruction rate for the “Toys” scene reaches a couple thousand fps with the RTX 3090 Ti and a couple hundred fps with the VU13P while both devices are limited by their computatioal power. For the other scenes, considering the amount of required memory to store all the variables, it is not possible to reach 1 fps with either device because of their limited bandwidth.

This study shows that the OR (One-Row model) reconstruction, for which reconstruction is performed line-by-line for the resulting image, can reach 25 fps for scenes up to 80×80×100 data cube size, i.e., image size times the number of bands, for actual GPUs and FPGA boards. However, the complexity of MR (Multi-Row model) is such that it can only be used for a reduced number of lines to fit memory constraints and available computing power.

### 5.6. Limitations

It is worth recalling that the estimation of the system performance we presented in this section is theoretical and assumes that we are able to exploit the computing devices at their full capacity. However, from our experience, we can state that it is better to consider 80% of the theoretical bounds as a conservative measure of performance due to the implementation on real devices. This is mainly due to slow-downs induced by data transfer latency. These slow-downs are difficult to evaluate without performing hardware simulations and are not accounted for in this theoretical study. Still, even when considering a 20% decrease in the performance, 34 fps is still achievable for a 200×200×33 scene. This is an encouraging result because state-of-the-art work proposed by Zhang et al. [38] achieved 6 fps for a 256×256×31 scene while using a dual camera, the DCCHI and a reconstruction method that combines the ADMM and the TWiST on a GPU implementation.

A second factor of limitations is the assumption that we have done on the model of the system matrix *H*. In this work, we assumed precise alignment of the optical system, in which the matrix of micro mirrors in the DMD is aligned with the pixels of the detector. When considering a real optical system, some misalignment might occur, or the information of a pixel emerging from the second prism might be spread over two pixels on the sensor. Correcting for these effects would require some further steps in the processing, leading to a potentially more complex reconstruction algorithm.

## 6. Conclusions

This paper addressed the feasibility of real-time hyperspectral CASSI reconstruction. From this perspective, two reconstruction methods [16,44] exploiting the compressed sensing and the snapshot feature of the DD CASSI system, proposed by [9], were evaluated from the criteria of quality of reconstruction, required computing power and required bandwidth, with a careful understanding of available trade-offs between the former and the others. These reconstruction methods, namely, the One-Row model and Multi-Row model, require some adaptations in order to achieve a significant reconstruction rate and reach real-time performance, both for a software target and a hardware custom-designed accelerator.

The first key aspect is to appropriately handle the very sparse nature of the system matrix used for iterative CGNE reconstructions. Some generic and custom-made data encodings are required to efficiently store the data to exploit their sparse structure and skip useless computations. Secondly, to reduce the power dissipation and increase the computing power, using fixed-point shows to be effective at a reasonable data format. Indeed, on the one hand, modern GPUs do allow the use of various set data formats, from 64-bit floating-point encoding down to some few-bit fixed points (4 bits), and on the other hand, the use of low complexity and finely tuned fixed-point operators in FPGA makes it possible to better exploit the available hardware and reach the highest performance. The design of a specific chip is also a third alternative that could benefit these results, although it was not deeply studied in this paper. Eventually, the balance in favor of one of these technologies highly depends on other considerations, and the provided method is expected to help a design team in its choices.

In conclusion, this paper proves that real-time reconstructions are possible, naturally at different rates with regard to the data cube resolution and hardware specifications. Simpler scenes can be reconstructed at a very satisfying rate, e.g., dozens of fps for a 200×200×33 scene, but increasing the spectral resolution is very demanding and puts a strain on performance, e.g., a dozen fps for a 100×100×100 scene, but the reconstruction rate can still be suitable for slower applications.

It is worth recalling that the analysis done in this study has been carried out considering the particular compressive hyperspectral camera DD-CASSI as a representative example of the CASSI group and the image reconstruction technique based on CGNE, which is a widely spread algorithm for solving inverse problems. Nevertheless, the general conclusions drawn in this paper are meant to still be valid (in terms of orders of magnitude for the performance and main trends) for other compressive devices (as they can be modeled by linear operators with sparse transfer matrices [14]) and other image reconstruction algorithms based on iterative variational approaches.

There are some possible further developments that would be interesting to tackle starting from this work, which are listed in the following. (i) Reproducing this analysis considering different CASSI devices; (ii) Considering alternative reconstruction algorithms such as accelerated versions of CGNE based on pre-conditioners, domain transforms, tests of different regularization terms and solvers based on deep learning; (iii) Designing an FPGA implementation of the CGNE by using high-level synthesis tools, which will give a more accurate estimation of the performance of the system.

## Figures and Tables

**Figure 1 sensors-22-09793-f001:**
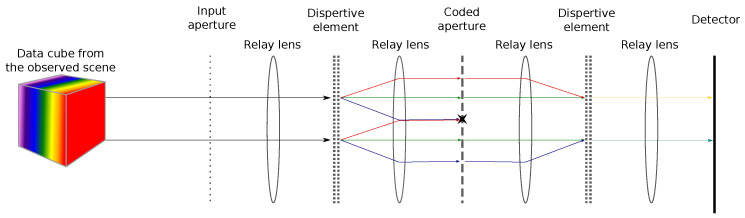
Principle of the DD CASSI system. Arrows represent the light rays that go through the system. Notice that because of the spectral multiplexing induced by the first dispersive element, light components from different light rays can be blocked by the same digit of the coded aperture. Hence, the figuratively yellow and indigo rays formed after the second dispersive element are missing some spectral components.

**Figure 2 sensors-22-09793-f002:**
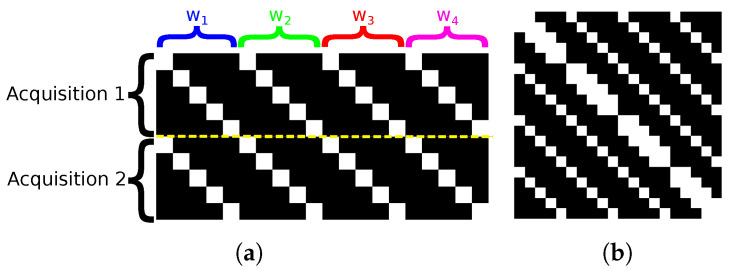
Graphical representations of matrices (**a**) ***H*** and (**b**) ***A*** matrices in the OR model, with N=2, C=5 and W=4. The white dots represent the non-zero entries. Here, the DMD is entirely in transmission mode to depict all the possible non-zero entries. The yellow dashed line separates the acquisitions.

**Figure 3 sensors-22-09793-f003:**
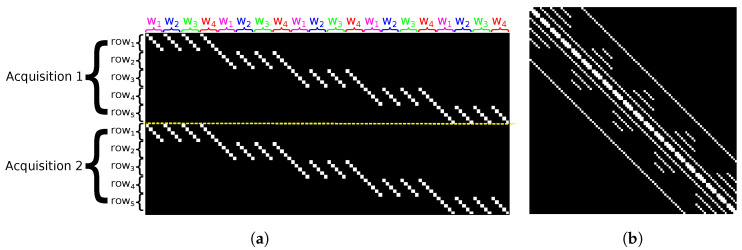
Graphical representations of matrices (**a**) ***H*** and (**b**) ***A*** in OR, with N=2, R=5, C=5 and W=4. The white dots represent the non-zero entries. DMD is entirely in transmission state to depict all the possible non-zero entries. The yellow dashed line separates the acquisitions.

**Figure 4 sensors-22-09793-f004:**
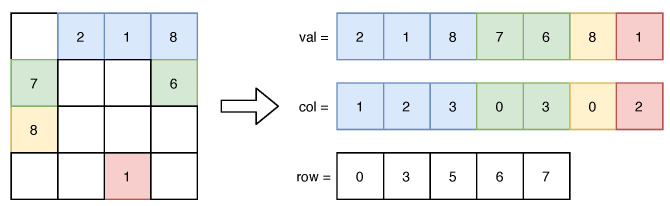
A dense matrix and its representation in CSR format. The row *i* is in the slice val[j:k], where j=row[i] and k=row[i+1]. The slice col[j:k] gives the column of the data.

**Figure 6 sensors-22-09793-f006:**
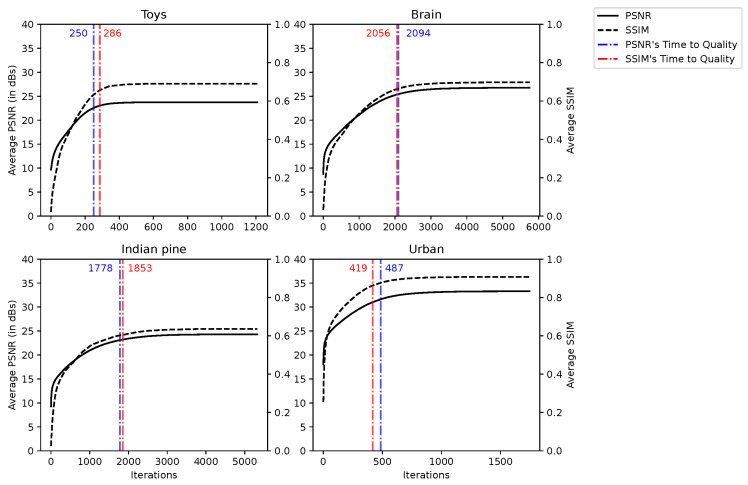
PSNR and SSIM for reconstructions of the CGNE over various iterations.

**Figure 7 sensors-22-09793-f007:**
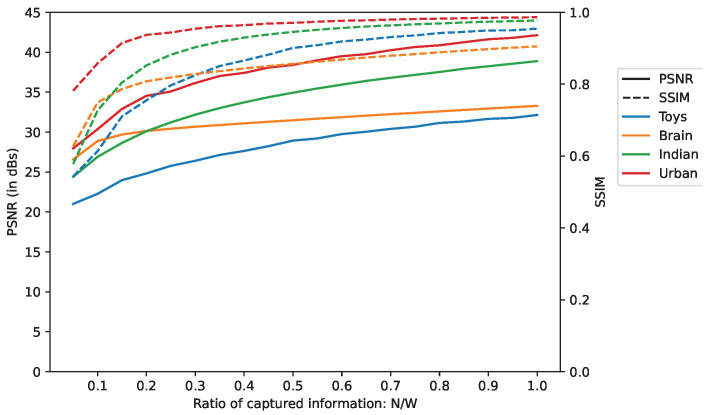
PSNR and SSIM depending on the quantity of captured information. *N* and *W* stand for the number of acquisitions and the spectral dimensions of the scene, respectively.

**Figure 8 sensors-22-09793-f008:**
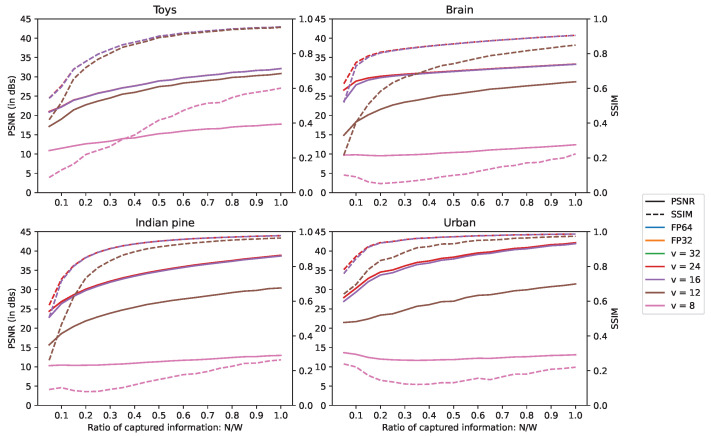
PSNR and SSIM depending on the quantity of captured information and data encoding. FP64 and FP32 denote, respectively, 64- and 32-bit floating-point data encoding; *v* is the number of bits of the the fixed-point’s fractional part.

**Figure 9 sensors-22-09793-f009:**
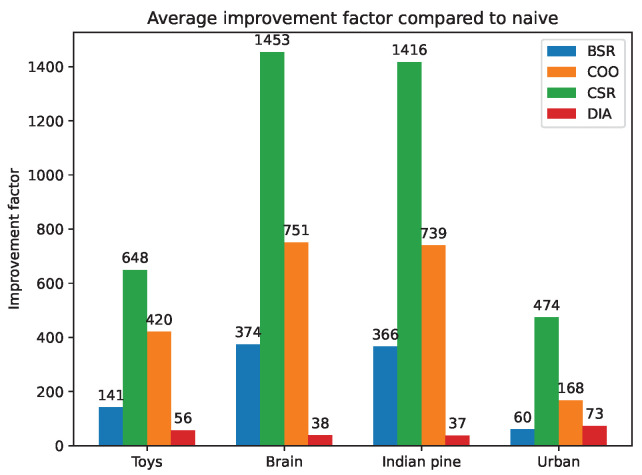
Processing time for CGNE reconstructions in the one-row model.

**Figure 10 sensors-22-09793-f010:**
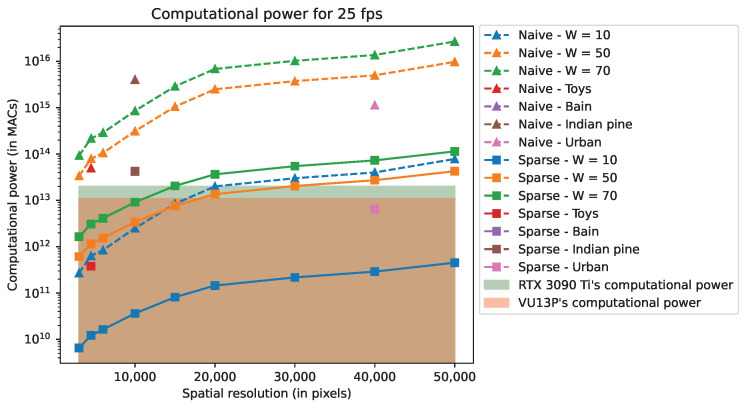
Computational power for 25-fps related to the RTX 3090 Ti and VU13P rated performances using the OR model.

**Figure 11 sensors-22-09793-f011:**
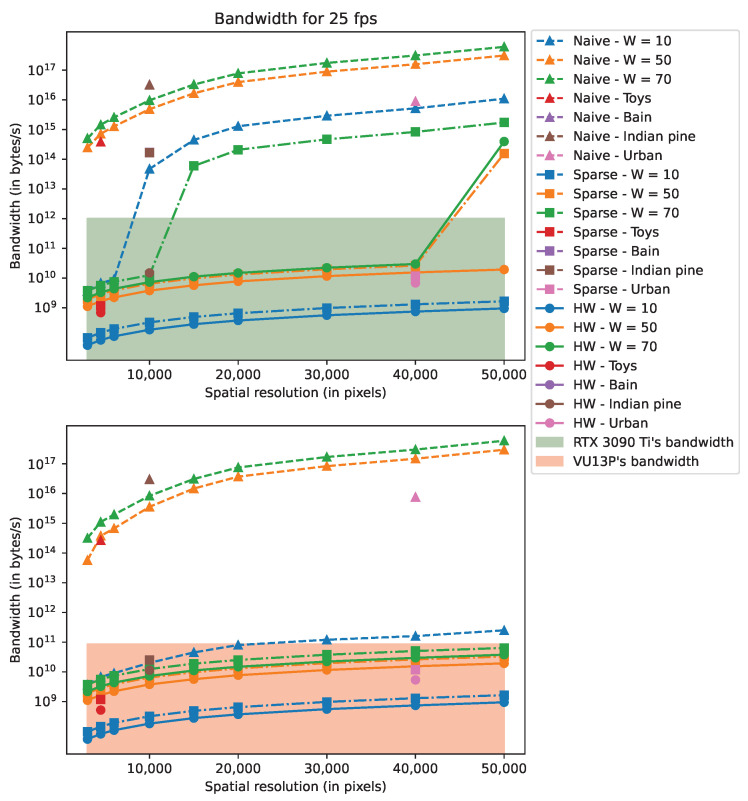
Bandwidth for 25-fps related to the RTX 3090 Ti and VU13P rated performances using the OR model.

**Table 1 sensors-22-09793-t001:** Selected devices with their computing power, work memory and data bandwidth.

Device	Nvidia RTX 3090 Ti	Xilinx Ultrascale + VU13P
Computing power	20.00 TMAC/s (CUDA cores)	10.95 TMAC/s
Work memory	6.114 MB	56.875 MB
Global memory	24 GB	128 GB
Global memory bandwidth	1008 GB/s	85.2 GB/s [47]

**Table 2 sensors-22-09793-t002:** Number of MAC and arithmetic operations needed for each step of Algorithm 1 considering the CSR format for A: *E* denotes the number of nonzero entries in A; see Equations (Equation 10) and (Equation 11).

Step	MAC	Arithmetic
1: $r_{0}=b-Ax_{0}$; $p_{0}=A^{⊤}r_{0}$	2E	RCW
3: $\alpha_{i}=r_{i}^{⊤}r_{i}/p_{i}^{⊤}p_{i}$	RCW	1
4: $x_{i+1}=x_{i}+\alpha_{i}p_{i}$;	RCW	0
5: $r_{i+1}=r_{i}-\alpha_{i}Ap_{i}$;	E+RCW	0
6: $\beta_{i}=r_{i+1}^{⊤}r_{i+1}/r_{i}^{⊤}r_{i}$	RCW	1
7: $p_{i+1}=A^{⊤}r_{i+1}+\beta_{i}p_{i}$	E+RCW	0

**Table 3 sensors-22-09793-t003:** Required computing power for one reconstruction and highest reconstruction rate depending on the computing device.

		Reconstructions per Second	
	Computational Cost per Scene Reconstruction	RTX 3090 Ti	VU13P
OR	15.15 GMAC	1320	723
MR	16.00 GMAC	1250	684

**Table 4 sensors-22-09793-t004:** CGNE’s memory footprint with fixed-point or 32-bit floating-point representation. “Single row” denotes the memory footprint for only one row in the OR model, and *v* denotes the number of bits for the fractional part of the fixed-point encoding.

	64-bit Floating-Point	Fixed-Point with *v* = 16 bits	32-bit Floating-Point
Single row	1.7 MB	0.76 MB	0.71 MB
OR	50.2 MB	22.8 MB	21.2 MB
MR	56.0 MB	27.2 MB	22.3 MB

**Table 5 sensors-22-09793-t005:** Bandwidth of a 25-fps video and maximum achievable fps with the OR model.

	Bandwidth for 25 fps	Max fps
RTX 3090 Ti	683 MB/s	36,890
VU13P	518 MB/s	4109

**Table 6 sensors-22-09793-t006:** Bandwidth of a 25-fps video and maximum achievable fps with the MR model.

	Bandwidth for 25 fps	Max fps
RTX 3090 Ti	2.06 TB/s	12
VU13P	626 MB/s	3402

**Table 7 sensors-22-09793-t007:** Final reconstruction rates in fps.

Scenes	RTX 3090 Ti	VU13P
Toys–(30×150×31)	1319.9	722.6
Brain–(100×100×100)	11.7	6.4
Indian pine–(100×100×100)	12.0	6.5
Urban–(200×200×33)	77.6	42.5

## Data Availability

No new data were created or analyzed in this study. Data sharing is not applicable to this article.

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
