# Peer review of "Feasibility of a Real-Time Embedded Hyperspectral Compressive Sensing Imaging System"

_sensors, 2022, doi:10.3390/s22249793_

Round 1

Reviewer 1 Report

This paper proposed an accelerated hyperspectral reconstruction method, which use a embedded device to present. The analysis presented the computation complexity of the method but also includes an estimation of memory footprint and bandwidth which are often overlooked in the literature but crucial when designing a device for real use. Extensive experimental results demonstrate the effectiveness of the proposed method. However, I have the following questions:

 1.      In Eq. (5), the authors can use Fourier transform to further accelerate the proposed reconstruction method.

2.      Some Hypersepctral compressive sensing reconstruction references are missing, such as Rank Minimization for Snapshot Compressive Imaging, λ-net: Reconstruct Hyperspectral Images from a Snapshot Measurement, Plug-and-Play Algorithms for Large-scale Snapshot Compressive Imaging, Snapshot compressive imaging: Theory, algorithms, and applications, Deep Tensor ADMM-Net for Snapshot Compressive Imaging, End-to-End Low Cost Compressive Spectral Imaging with Spatial-Spectral Self-Attention, Snapshot compressed sensing: performance bounds and algorithms, Learning Nonlocal Sparse and Low-Rank Models for Image Compressive Sensing, A benchmark for sparse coding: When group sparsity meets rank minimization, Group sparsity residual constraint with non-local priors for image restoration.

Reviewer 2 Report

This paper provides an in-depth analysis of the required performance in terms of computing power, data memory and bandwidth considering a compressive hyperspectral imaging system. The results of the analysis show that real-time is possible by combining several approaches, namely the exploitation of the system matrix’s sparsity and bandwidth reduction by appropriately tuning the data value encoding. This paper would promote the practical usage of CASSI. It would be better to test the performance with a physical CASSI system. 

Reviewer 3 Report

The manuscript showed feasibility of a real-time embedded hyperspectral compressive sensing imaging system.How uncertain is this approach? What are its limitations? Has this approach been used in other scientific studies? Are there any alternative approaches?

The literature review mainly focuses on the work carried out in this paper, and the review of other algorithms is not enough.

Section2,3,4 are materials and method.The structure is not clear enough. It is suggested to write according to the innovation points of this article.

The key assumptions and major limitations need to be clarified. 

How does this study contribute to existing literature on the study topic? You must compare your findings with literature. 

Round 2

Reviewer 3 Report

This paper is a prospective study which brings out ways to accelerate the hyperspectral reconstruction method and determine its performance on a embedded device. The approaches improve the performance on both software and hardware aspects. Please the formats of the tables and the formula.

Author Response

Thanks for your remark.

We have double checked the format of tables and equations in the manuscript.

In the revised manuscript we have improved the layout of equations 12 and 13. We have modified the format of Table 1, 2, 4 and 7.